# An Overview on Cyclic Fatty Acids as Biomarkers of Quality and Authenticity in the Meat Sector

**DOI:** 10.3390/foods9121756

**Published:** 2020-11-27

**Authors:** Veronica Lolli, Emanuela Zanardi, Aidan P. Moloney, Augusta Caligiani

**Affiliations:** 1Department of Food and Drug, University of Parma, 43124 Parma, Italy; emanuela.zanardi@unipr.it (E.Z.); augusta.caligiani@unipr.it (A.C.); 2Teagasc, Animal & Grassland Research and Innovation Centre, Grange, Dunsany, Co., C15 PW93 Meath, Ireland; aidan.moloney@teagasc.ie

**Keywords:** cyclopropane fatty acids, ω-cyclohexyl fatty acids, fat, animal diet, species, meat quality, GC-MS, ^1^HMNR, food authenticity

## Abstract

A survey was conducted to determine the content of cyclopropane fatty acids (CPFAs) and ω-cyclohexyl fatty acids (CHFAs) by using gas chromatography- mass spectrometry (GC-MS) and proton nuclear magnetic resonance (^1^H NMR) techniques in various meat samples from different species, including commercial samples and complex and thermally processed products (i.e., Bolognese sauce). The CPFAs concentration (as the sum of two isomers, namely dihydrosterculic acid and lactobacillic acid) in bovine meat fat (ranging between 70 and 465 mg/kg fat) was positively related to a silage-based diet, and therefore, they are potential biomarkers for monitoring the feeding system of cattle. CHFAs, such as 11-cyclohexylundecanoic and 13-cyclohexyltridecanoic acids, were only found in lipid profiles from ruminant species, and a linear trend was observed in their content, together with *iso*-branched fatty acids (*iso*-BCFAs) deriving from ruminal fermentation, as a function of bovine meat percentage in both raw and cooked minced meat. Thus, CHFAs are potential biomarkers for the assurance of the meat species and, combined with *iso*-BCFAs, of the beef/pork ratio even in complex meat matrices. The proposed approaches are valuable novel tools for meat authentication, which is pivotal in the management of meat quality, safety, and traceability.

## 1. Introduction

Traceability of meat has become more important in recent years with the globalization of food distribution combined with the potential for fraudulent claims on commercial products. In the meat sector, the substitution of high-cost meats with low-priced alternatives, especially for those designated as Protected Designation of Origin (PDO) or Protected Geographical Indication (PGI), and the mislabeling of meat species are significant issues [1,2].

Different analytical methods (e.g., liquid- and gas chromatography (L/GC), mass spectrometry (MS), Raman spectroscopy, and low-field nuclear magnetic resonance (NMR) and near-infrared (NIR) spectroscopy) are available for the authentication of meat and meat products in terms of species identification and dietary patterns [2,3,4,5,6].

Several authors have discussed the identification of meat from different species using analytical techniques based on DNA analysis, traditional and real-time polymerase chain reaction (PCR), and protein/peptides measurements [7,8,9]. 

The fatty acid composition of muscle and adipose tissue is less reliable for meat species identification because this parameter is largely dependent on the animal diet [10]. Besides diet, numerous other factors also determine the quality of tissue fats. For example, the digestion process of dietary lipids (such as ruminal hydrolysis or bypass of protected fat in ruminants and degree of lipolysis in monogastrics) and their re-esterification in the gut endothelium may significantly differ among animal species. Furthermore, the nutrient demand of different species is primarily different, and it reflects in the composition of the stored fats [11,12].

A possible approach for meat species identification based on the positional distribution of fatty acids in triacylglycerols and 2-monoacylglycerols of adipose tissue was proposed by Szabó et al. (2007) [13], who tested subcutaneous fat samples of red deer, moose, wild boar, extensively farmed pig, badger, rabbit, and goose.

In beef, several components have been proposed for the authentication of dietary history. Markers of plant feeding, such as carotenoid pigments, n-3 polyunsaturated fatty acids (n-3 PUFA), specific conjugated linoleic acid (CLA) and C18:1 isomers, ascorbic acid, α-tocopherol, terpenes, sulfur compounds, phenols, and ratio of carbon stable isotopes, were shown to be valuable tools for distinguishing between grass and concentrate-fed ruminants [14,15]. However, most beef production systems are not solely based on grass or grain but consist of varying proportions of each at different stages of production. Therefore, the real application or value of the proposed components will rely on their ability to distinguish between animals in a commercial setting.

For example, providing ensiled forages is forbidden during the last four months of the fattening period by the specifications of the of PGI European mark of the “Vitellone Bianco dell’Appennino Centrale” (Reg. EU n. 134/98). Moreover, grass-fed beef is more appreciated by consumers due to its “green-healthy image” [16] and a perception of a higher nutritional quality and of being produced within a more animal welfare-friendly and environmentally sustainable production system.

Previous data [17] indicated that meat fatty acid composition could discriminate between ruminant feeding systems based on grazed grass and/or concentrate. However, information is lacking on a specific analytical marker to distinguish between beef from grazed or grass silage-fed cattle.

Cyclopropane fatty acids (CPFAs), such as dihydrosterculic and lactobacillic acids, and ω-cyclohexyl fatty acids (CHFAs), such as 11-cyclohexylundecanoic acid (ω-11 CHFA) and 13-cyclohexyltridecanoic acid (ω-13 CHFA), are microbial alicyclic fatty acids that occur in lactic acid bacteria and in rumen microorganisms, respectively [14], and were recently detected in foods of animal origin [18,19,20].

The detection of CPFAs is a growing field of research in food lipids, both in food characterization/authentication and in food safety and nutrition, and different analytical methods for their determination have been developed [21,22]. These studies suggest the presence of CPFAs as minor compounds of the lipid profile, especially of dairy and meat products, and indicated a positive relationship between their presence in these products and the use of ensiled feed. In particular, in the dairy sector, a GC-MS qualitative and quantitative methodology was developed and validated (UNI11650) to determine CPFAs in cheese fat [21], which is currently used for the authentication of Parmigiano Reggiano cheese (confirming the absence of silage in the cows ration, which is forbidden by the product specifications). Moreover, preliminary results from NMR analysis [22] showed potential for measuring CPFAs in meat to confirm the absence of silage in the ration of animals from which beef is labelled with a PGI, which prohibits silage feeding. 

With regard to the other class of cyclic fatty acids, CHFAs were detected in cow milk [18] and in the meat of ruminant species, specifically bovine and ovine meat, together with *iso*-branched fatty acids (*iso*-BCFAs) [23].

Previous studies [18,24] suggested that both CHFAs and *iso*-BCFAs in dairy foods mainly originate from bacteria leaving the rumen. Recent experiments [23] showed the possibility of combining these compounds to discriminate beef from pork in both raw and cooked minced beef. For this purpose, this study focused on the identification and characterization of cyclic fatty acids in a large number of meat samples, with the aim to explore the possibility of using CPFAs to discriminate between beef from grass silage-based diets and grass-based diets, and CHFAs (together with *iso*-BCFAs) as markers of ruminant species to detect and quantify the beef/pork ratio in commercial minced meat.

## 2. Materials and Methods

### 2.1. Chemicals

Methanol, n-hexane, dichloromethane, deuterated chloroform, anhydrous sodium sulphate, and tetracosane (purity 99%) were obtained from Sigma-Aldrich (Saint Louis, MO, USA) and potassium chloride and potassium hydroxide pellets were obtained from Carlo Erba (Milan, Italy). Dihydrosterculic acid methyl ester pure standard was obtained from Abcam (Cambridge, UK). All the performed chemicals as solvents, standards, and reagents were of analytical grade.

### 2.2. Sampling

More than one hundred different meat samples were investigated for CPFAs and CHFAs. Specifically:‒Batch 1: commercial meat samples.

Fifty meat samples were obtained from a local market (Parma, Italy), including 18 samples of fresh beef meat and 32 samples of both fresh and cured meat products from other species: horse (*n* = 8), rabbit (*n* = 3), pork (*n* = 5), cured pork meat (*n* = 6), lamb (*n* = 2), goose (*n* = 2), and chicken/turkey (*n* = 6).

Selvedge fat was removed from each cut of meat and stored at −20 °C until fatty acid analysis. The remaining meat (lean cut) of about 200 g was finely minced using a commercially available electric mincer (Moulinex Moulinette Compact DJ3051). 

Six calibration mixtures of both raw and cooked minced meat (i.e., Bolognese sauce) from a previous study [25] were used, mainly to optimize the GC-MS method performed for CHFAs and *iso*-BCFAs determination in meat fat. Specifically, they were prepared with a known meat composition, ranging from 100% to 0% beef, at a pilot plant scale of an Italian food company by performing a simulated industrial thermal processing of commercial Bolognese sauce (cooked minced meat). Two test (unknown) samples of Bolognese sauce with different ratios of beef/pork were also provided by the same pilot plant scale to verify the applicability of the GC-MS method.
‒Batch 2: certified meat samples for the feeding system.

Twelve beef meat samples (*Longissimus dorsi* from Limousin, Charolais, or crossbred animals aged 17–23 months) were kindly provided by Prof. Riccardo Bozzi (University of Florence, Florence, Italy) and certified for the absence of ensiled feeds in the animal diet.

Forty-five beef samples (cube roll steak) from Irish suckler bulls (aged 15 months) from a previous study [16] and belonging to three certified dietary groups (*n* = 15) were provided: (a) grass silage ad libitum plus 5 kg of concentrate (SC); (b) grazed grass without supplementation (G0); (c) grazed grass plus 0.5 kg of the dietary dry matter intake as concentrate (GC).

### 2.3. Fat Extraction

Lipid extraction was carried out following the method of Folch et al. (1957) [26], slightly adapted [22]. Precisely, samples of both fat and minced muscle (2 g) were mixed with 50 mL of dichloromethane:methanol (2:1, *v/v*) for 30 min, centrifuged at 3000 rpm for 10 min, and filtered. This procedure was repeated two times. The two filtrates were collected and added to a volume of about 35 mL of KCl 0.88% in distilled water. The mixture was shaken vigorously in a separatory funnel. The final biphasic system was decanted and the lower organic phase was collected, dried with anhydrous sodium sulphate, and filtered. Finally, the extracted fat was recovered after solvent was evaporated under a vacuum by using a rotary evaporator.

### 2.4. GC-MS Analysis

GC-MS analysis was performed on an Agilent Technologies 7820A gas chromatograph (Agilent Technologies, Palo Alto, CA, USA) coupled to an Agilent Technologies 5977B single quadrupole mass spectrometer, as previously reported by Caligiani et al. (2016) [21]. Briefly, 100 mg of the lipid extract was dissolved in hexane (5 mL) containing the internal standard tetracosane at 0.05 mg/mL and vortexed for 1 min with 0.2 mL of 10% KOH/MeOH. After phase separation, the superior organic phase was recovered and 1 µL was injected into the GC-MS system (split mode 20:1, split flow 19.6 mL/min). A low-polarity capillary column (SLB-5ms, 30 m, 0.25 mm, 0.25 µm, Supelco, Bellafonte, PA, USA) was used. The mass spectrometer operated in the electron impact (EI) ionization mode (70 eV) and the ion source temperature was set at 230 °C. Data acquisition was done in the scan mode (40–500 *m/z*) with a programmed temperature from 60 to 280 °C.

The relative concentrations (as the mean of three replicates) of the fatty acid methyl esters (FAMEs) of interest (CPFAs and CHFAs and *iso*-BCFAs) were calculated by manually integrating their peak areas, with respect to the internal standard tetracosane. Finally, concentrations were reported as mg/kg of total fat.

Specifically, CPFAs identification in meat samples was performed by GC-MS, based on the mass spectrum previously published for lactobacillic acid methyl ester (M^+^: *m/z* 310; characteristic ion *m/z* 278) [18] and on the reference library (NIST 11), and then compared with the mass spectrum and retention time of dihydrosterculic acid methyl ester standard calibration solution (Figure 1). Finally, ^1^H NMR analysis was performed in order to confirm the cyclopropane ring presence in meat fat and to correctly quantify CPFAs (see below). Positivity or negativity of samples for CPFAs was based on the threshold of 60 mg/kg of fat according to Caligiani et al. (2016) [21].

CHFAs identification was conducted by referring to the mass spectrum (Figure 1) and retention time (SLB-5 ms column) previously published by Marseglia et al. (2013) [18]: M+ *m/z* 282 for ω-11 CHFA (which elutes together with the oleic acid (C18:1) methyl ester) and M+ *m/z* 310 for ω-13 CHFA (which elutes between eicosenoic acid (C20:1) and arachidic acid (C20:0) methyl esters). A semi-quantification of the characteristic molecular ions of both CHFAs was performed with respect to the molecular ion *m/z* 85 of the internal standard tetracosane.

### 2.5. ^1^H NMR Analysis

^1^H NMR analysis was performed according to Lolli et al. (2018) [22]. For this analysis, 100 mg of meat fat was accurately weighed and dissolved in 1 mL of CDCl_3_ containing 0.01 mg of trimethylsilyl decanol (TMSD) as an internal standard.

^1^H NMR spectra were acquired on a VARIAN INOVA-600MHz spectrometer (Varian, Palo Alto, CA, USA), equipped with a 5-mm triple resonance inverse probe. Data were collected at 298 K, with 32 K complex points, using a 90° pulse length. In total, 1024 scans were recorded with an acquisition time of 1.707 s and a recycle delay of 2 s. Pre-saturation of the fatty acids’ CH2 signal (1.25 ppm) was applied. MestReC software 6.0.2 (Santiago de Compostela, Spain) was utilized for processing the NMR spectra, which were Fourier-transformed with an FT size of 64 k and a 0.2-Hz line-broadening factor, phased and baseline-corrected, and referenced to the chloroform signal (7.26 ppm).

### 2.6. Data Analysis

Concentration values (mg/kg total fat) are reported as the mean ± standard deviation (SD) of all samples in a particular group of each batch.

The association between CHFAs and *iso*-BCFAs concentrations (as an average of three replicates) and the beef percentage in minced meat was examined using Pearson’s correlation coefficient (r), and a quadratic regression model equation (y = ax^2^ + bx + c) was performed to predict the beef/pork ratio in the two test blind samples of Bolognese sauce from Batch 2.

All the analyses were performed with Microsoft Office Excel 2013 Analysis ToolPak.

## 3. Results

### 3.1. CPFAs Determination in Meat Fat

The presence of CPFAs in meat samples was analyzed by combining GC-MS and ^1^H NMR, which evidenced the characteristic signal of the cyclopropane ring (at −0.35 ppm) [27] used for CPFAs quantification [22].

Regarding samples from Batch 1, CPFAs were found in all the commercial beef meat samples (ranging between 70 and 410 mg/kg of total fat) but not in the meat of other species (i.e., horse, rabbit, pork, cured pork meat, lamb, goose, and chicken/turkey). No significant difference (*t*-test, *p* > 0.05) was found between the CPFAs amount in bovine adipose tissue lipids and intra-muscular fat (data not shown), thus suggesting that CPFAs are equally distributed.

These CPFAs were also detected in samples from Batch 2 (of certified origin), especially in all meat samples from cattle having an ensiled feed-based diet (SC) with concentrations ranging between 108 and 465 mg/kg of total fat. 

Muscles from two bulls from the grazing group (G0) and muscles from seven bulls from the grazing plus concentrates group (GC) had detectable levels of CPFAs (results confirmed by ^1^H NMR analysis). However, these animals were fed silage before the beginning of grazing (and a small amount of concentrate) [16]. Therefore, these data seem to suggest a “carry over” of CPFAs when cattle had stopped consuming silage. On the other hand, most of the certified meat samples from cattle not fed silages (only grazed grass and concentrate) were negative for CPFAs.

Data obtained from the quantitative analysis are summarized in Table 1.

### 3.2. Omega-Cyclohexyl Fatty Acids Determination in Meat Fat

A semi-quantitative GC-MS method was used to determine CHFAs in several meat samples from different species. The experimental results obtained in this work combined with our previous preliminary data [23] suggested the presence of CHFAs, both ω-11 CHFA and ω-13 CHFA, only in meat from ruminants (see above), with values in the range of 90–230 and 20–200 mg/kg of the total fat, respectively. Table 2 shows the positive correlation between the concentration (mg/kg total fat) of both CHFAs and *iso*-BCFAs with bovine meat percentage. 

Furthermore, the response pattern of the GC-MS analysis related to CHFAs and *iso*-BCFAs, both in raw and processed minced meat, has been further examined and a linear trend (R^2^) was observed as a function of bovine meat percentage, as shown in Figure 2.

Specifically, in fresh meat mixtures, linearity was observed for these compounds in the range of 0–100% bovine, except for ω-11 CHFA, for which linearity was determined in the range of 40–100% bovine as its concentration is under its limit of detection for this matrix.

Bolognese sauce is a more complex matrix than fresh minced meat and CHFAs could not be detected in mixtures under 60% beef (Figure 2). On the contrary, a good linear trend and a good sensitivity were observed for *iso*-BCFAs in the range from 0% to 100% beef for these complex mixtures.

Therefore, based on these observations, we explored the possibility of using the quantification of *iso*-BCFAs to predict bovine/pork meat percentage in test blind samples of Bolognese sauce with different ratios of beef/pork. With this aim, quadratic regression was used as a model, as it best fitted this set of data. In particular, for the percentage of beef, the following equations were applied: (a) (*iso*-C16:0): y = 0.039x^2^ + 4.01x + 48.60R^2^ = 0.97; (b) (*iso*-C17:0): y = 0.07x^2^ + 4.20x + 67.79R^2^ = 0.98). Results of the trueness of beef/pork percentage estimations are shown in Table 3.

## 4. Discussion

### 4.1. Application of CPFAs in Meat as Markers of Animal Feed Regimen

Preliminary data on a small number of samples obtained by combining GC-MS and ^1^H NMR analysis, and previously discussed by Lolli et al. (2018) [22], suggested that CPFAs were present in fresh commercial beef meat samples and not in pork and chicken meat. In addition, CPFAs were not detected in two samples of certified meat from cows not fed fermented forages. In this paper, the investigation of the presence of CPFAs was extended to almost 100 other meat samples, comprising commercial beef meat, certified meat from different feeding practices (including or not including silages), and meat from other species. Above all, the results from Batch 2 reflect a positive relationship between the consumption of silages and the presence of CPFAs in the fatty acid profile, previously observed for dairy products [18,19,21], thus demonstrating their potential use to discriminate between beef from grass silage-based diets and grass-based diets, even in the meat sector.

The relationship between silage intake and their permanence in tissues would be a point of interest for monitoring the period of time over which a non-silage ration is required, according to certain product specification rules (from PDO and PGI status). For example, due to the possible negative effects on meat quality (such as meat color and water-holding capacity), the productive rules of “Vitellone Bianco dell’ Appennino Centrale” forbid the use of maize silage in the four months before slaughtering [28]. 

Finally, further detailed studies are needed to elucidate the transfer rate of CPFAs from daily forage ratios and distribution in animal tissues together with their biological significance.

### 4.2. Omega-Cyclohexyl Fatty Acids: Species Identification and Quantitation of the Ratio of Beef to Pork in Minced Meat

The experimental results obtained in this work supported previous data [23] suggesting the presence of ω-11 CHFA and ω-13 CHFA only in meat from ruminant origin, in both raw and processed products, such as Bolognese sauce. Data suggest that CHFAs and *iso*-BCFAs levels are proportional to the increase in beef percentage in both raw and processed meat.

Furthermore, from a quantitative point of view, the beef/pork percentage ratio could be predicted by quadratic regression from the concentrations of *iso*-BCFAs, thus, in a complex matrix after cooking (i.e., simulated industrial thermal processing for cooking commercial Bolognese sauce, reaching a temperature of about 90 °C [25]).

The concentration of beef in the mixture was a little underestimated, but trueness was always within 86–99% of the nominal value for beef. Moreover, the presence of pork could be detected at 20% and adulteration behind this value may have limited commercial significance.

In general, CHFAs can distinguish beef from pork, especially from a qualitative point of view, even after thermal treatment (i.e., cooking) and in processed products for which visual inspection is more difficult (i.e., minced meat). As mentioned above, current analytical methods for species discrimination in meat products are mainly based on protein or DNA measurement, which are not directly comparable to labelled meat expressed as percentage (*w/w*) [2,25]. Furthermore, analytical procedures based on protein analysis are sensitive to heat treatment. As such, CHFAs are molecular biomarkers of ruminant meat and their detection combined with the quantitation of other ruminal fatty acids, such as *iso*-BCFAs, could enforce current analytical methods applied for labelling regulations. 

## 5. Conclusions

In the present study, we explored the potential use of the carboalicyclic fatty acids CPFAs and CHFAs as biomarkers for authenticity and to protect against food fraud in the meat sector.

The results indicated that the presence of CPFAs detected in bovine meat (ranging between 70 and 465 mg/kg fat) is directly associated with the use of ensiled forages. This could be used to distinguish between silage-based diets and grass-based diets, particularly to ensure compliance with quality schemes, such as the PGI, whose producers declare and certify the absence of silages in the ration of cattle. Further work is in progress to test the presence of CPFAs in different ensiled feeds to expand this relationship.

Regarding CHFAs, the results supported their use, combined with *iso*-BCFAs, as biomarkers of ruminant species for quality control and detection of meat adulteration, especially mislabeling and species substitution. The latter is of paramount importance in terms of food safety because adulteration may lead to changes in exposure to hazards as well as loss of traceability and an increase in associated risks.

## Figures and Tables

**Figure 1 foods-09-01756-f001:**
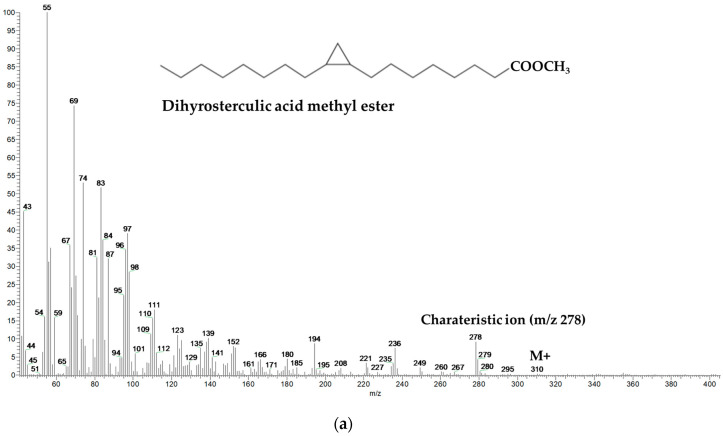
Mass spectra and chemical formulas of cyclic fatty acids detected in meat fat as methyl esters: (**a**) dihydrosterculic acid methyl ester (M+: *m/z* 310; characteristic ion fragment *m/z* 278); (**b**) ω-11 cyclohexylundecanoic acid methyl ester (M+: *m/z* 282); (**c**) ω-13 cyclohexyltridecanoic acid methyl ester (M+: *m/z* 310).

**Figure 2 foods-09-01756-f002:**
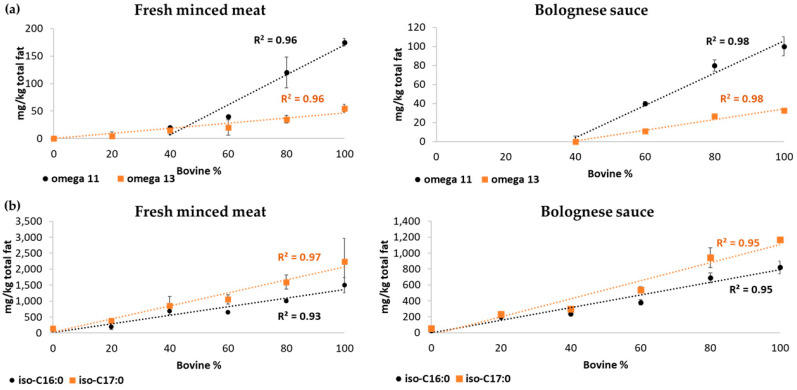
Linear trend observed for (**a**) CHFAs (ω-11 CHFA (black) and ω-13 CHFA (orange)) and (**b**) *iso*-branched fatty acids (BCFAs) (*iso*-C16:0 (black) and *iso*-C17:0 (orange)) concentrations in fresh minced meat or in Bolognese sauce with different ratios (0–100%) of beef and pork.

**Table 1 foods-09-01756-t001:** Presence of cyclopropane fatty acids (CPFAs) ^1^ in meat samples.

Samples	No. Positive ^2^ for CPFAs	Mean ± SD (mg/kg fat)	Range (mg/kg fat)
Batch 1			
Commercial beef meat ^3^	18/18	200 ± 100	70–410
Other meat products ^4^	0/32	<LOD ^5^	<LOD ^5^
Batch 2			
*Longissimus dorsi* ^6^	2/12	240 ± 57	200–280
G0 ^7^	2/15	164 ± 23	147–180
GC ^8^	7/15	158 ± 75	79–268
SC ^9^	15/15	240 ± 120	108–465

^1^ CPFAs concentrations are reported as the sum of dihydrosterculic acid and lactobacillic acid [21]; ^2^ positive for CPFAs means a concentration (≥60 mg/kg total fat) and was experimentally confirmed by ^1^H NMR analysis [22]; ^3^ fresh beef meat from a local supermarket (Parma, Italy); ^4^ fresh commercial meat (horse, rabbit, pork, cured pork meat, lamb, goose, and chicken/turkey) from a local supermarket (Parma, Italy); ^5^ LOD = limit of detection (<60 mg/kg total fat) [21]; ^6^ from Limousin, Charolais, or cross-bred animals aged 17–23 months and certified for the absence of ensiled feeds in the dietary; ^7^ G0: fed only grazed grass; ^8^ GC: fed grazed grass plus concentrate; ^9^ SC: fed grass silage plus concentrate.

**Table 2 foods-09-01756-t002:** Pearson’s correlation coefficient (r) between cyclohexyl fatty acids (CHFAs) or *iso*-branched fatty acids concentrations and the percentage of bovine meat both in raw and cooked minced meat.

	Pearson’s r	
Fatty Acids	% Bovine (Raw Meat) ^1^	% Bovine (Bolognese Sauce) ^1^
ω-11 CHFA	0.93	0.95
ω-13 CHFA	0.97	0.93
*iso*-C16:0	0.97	0.98
*iso*-C17:0	0.98	0.98

^1^ Six calibration mixtures of beef and pork meat ranging from 100% to 0% beef.

**Table 3 foods-09-01756-t003:** Beef and pork ratio calculations in the two blind samples of Bolognese sauce. Data are presented as the average of three independent replicates ± standard deviation. The trueness ^1^ values are reported in brackets.

			*iso*-C16 ^2^		*iso*-C17 ^3^	
Test Sample	Real ^3^ Beef %	Real ^3^ Pork %	Calculated Beef %	Calculated Pork ^4^ %	Calculated Beef %	Calculated Pork ^4^ %
A	35	65	30.9 ± 3.1 (88%)	69.1 ± 3.1 (106%)	30.2 ± 2.3 (86%)	69.8 ± 2.3 (107%)
B	75	25	74.1 ± 0.2 (99%)	25.9 ± 0.2 (104%)	70.4 ± 2.1 (94%)	29.6 ± 2.1 (118%)

^1^ Trueness refers to the percentage obtained from the ratio between calculated percentage from linear regression and the actual percentage of bovine and pork in the tested sample; ^2^ estimation was performed using linear regression from *iso*-C16:0 quantification in calibration mixtures of Bolognese sauce from 100% to 0% beef; ^3^ estimation was performed using linear regression from *iso*-C17:0 quantification in calibration mixtures of Bolognese sauce from 100% to 0% beef; ^3^ Real refers to the actual percentage of bovine and pork in the samples (nominal value); ^4^ percentage obtained by difference.

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
