# Peer review of "An Overview on Cyclic Fatty Acids as Biomarkers of Quality and Authenticity in the Meat Sector"

_foods, 2020, doi:10.3390/foods9121756_

Round 1

Reviewer 1 Report

The paper entitled "An overview on the detection of cyclic fatty acids as biomarkers of quality in the meat sector"
by Lolli et al. deals with determination of the content of cyclopropane (CPFA) and ω-11 cyclohexyl fatty acids (CHFA) by
GC-MS and 1HNMR in various meat samples from different species, including complex and thermally processed products
The CHFA were proposed as potential biomarkers for the assurance of the meat species and combined with iso-BCFA of the
beef/pork ratio, also in complex meat matrices. The results indicated that the presence of CPFA detected in bovine meat
is directly associated with the use of ensiled forages. This could be used to distinguish between silage-based diets and
grass-based diets. The proposed approaches are presented as valuable novel tools for meat authentication.

The article is well written and provides valuable results potentially useful from the practical point of view.

Author Response

Authors thank for the comments of the Reviewer.

Reviewer 2 Report

There are some minor comments that I have for the authors - use italics for 'iso'; i.e. iso-17:0 as an example and, at line 193, use 't-test' rather than 'T-test'.

One concern that I have for the plots, is that I am not convinced that the use of linear regression to describe the plots is not appropriate. I believe that quadratic regression (i.e. y = a0 + b1x + b2x2) would be better used as a model. I would strongly encourage the authors to review this approach.

Author Response

Author thank for the comments of the Reviewer.

Authors revised the Manuscript according to the suggestions of the Reviewer. Specifically:

  • italics for 'iso' has been used throughout the text.
  • 'T-test' has been replaced with 't-test' (at line 191).
  • Regarding this comment: "One concern that I have for the plots, is that I am not convinced that the use of linear regression to describe the plots is not appropriate. I believe that quadratic regression (i.e. y = a0 + b1x + b2x2) would be better used as a model. I would strongly encourage the authors to review this approach".

    Since data suggested that CHFA and iso-BCFA levels were direct proportional to the increase of beef percentage in both raw and processed meat, author decided to keep the linear function to describe this trend from a qualitative point of view. For CHFA, linearity is observed over the range of the limit of detection (LOD): LOD was ≤ 60% beef in the mixtures of Bolognese sauce and especially for omega 11 CHFA, LOD was ≤ 40% beef in fresh mixtures. On the other hand, from a quantitative point of view, quadratic regression from the concentrations of iso-BCFA was applied as a model as it best fitted for predicting beef/pork ration in calibration mixtures of Bolognese sauce. Authors revised Figure 2 (Figure 1 at line 228) and the overall text according to these new considerations.

Reviewer 3 Report

The authors have explored the potential use of cyclic fatty acids such as cyclopropane fatty acids, ω-68 cyclohexyl fatty acids and iso-branched fatty acids for the authentication of meat and meat products. Previously published GC-MS and 1H-NMR methods have been used to analyze various meat samples and the statistical evaluation of the analytical data led to important conclusions on the potential use of these biomarkers.

Overall, the research theme and the experimental work presented in this manuscript are interesting, the manuscript is well-organized.

Comments:

  1. Acronyms should be annotated once and then used: Line: 11 cyclopropane (CPFA) / Line 68: Cyclopropane fatty acids (CPFA) and Line 12 ω-11 cyclohexyl fatty acids (CHFA)/ line 69: ω-68 cyclohexyl fatty acids (CHFA)
  2. The chemical structures of all the cyclic fatty acids tested, could be included in a figure
  3. Line 206: A semi-quantitative GC-MS method was optimized?, please either report the optimization steps or replace “optimized” by “used”.
  4. A Figure with representative GC-MS chromatograms like in Figure 1 (1NMR procedure), could be included

Author Response

Author thank for the comment of the Reviewer. Author modified the text according to his/her suggestions. Specifically:

  1. Acronyms at lines 68-69 have been used according to the suggestion of the reviewer.
  2. The chemical structures of all the cyclic fatty acids tested are in the graphical abstract
  3. Line 206: the term “optimized” has been replaced by “used”.
  4. Author decided to delete Figure 1 and to depict overall analytical results  in the graphical abstract